# Interpreting Neural Networks to Understand Written Justifications in Values-Affirmation Essays

## Abstract

We report efforts to interpret a Long-Short Term Memory neural network trained to recognize gender and writing instructions on a set of essays from a psychological educational intervention known as a values affirmation. Adjusting the model at test time to output sequential probabilities as each new token is encountered, rather than predicting the class holistically, we query the model with carefully constructed sentences designed to test a theoretically informed hypothesis: male versus female students write in a way that reflects a greater emphasis on independence versus interdependence, respectively. The LSTM model outperforms the baseline, and the model's predictions to our constructed test sentences suggest modest support for these hypotheses.

## 1 Introduction

The difficulty of interpreting neural networks is a major barrier to their wider adoption among applied quantitative scientists. Although these models have achieved a high level of performance across a number of NLP and vision processing tasks (Sutskever et al., 2014; Bahdanau et al., 2014; Socher et al., 2011; Chen and Manning, 2014), attempts to understand why a model performs as it does have been less frequent. Some recent work in this area has been devoted to understanding long-term dependencies such as bracketing (Hermans and Schrauwen, 2013; Karpathy et al., 2015), studying learned sparse word vectors (Faruqui et al., 2015) and to generating rationalization of predictions without manual annotations (Lei et al., 2016). These efforts are important for understanding where and why a model ex-

periences shortcomings. Additionally, as we address here, finding straightforward ways to interpret these models will increase their applicability across the sciences more generally.

In this paper, we take a somewhat different approach to understanding properties learned by a Long-Short Term Memory (LSTM) network (Hochreiter and Schmidhuber, 1997). Rather than exploring the components of the model itself, we take a hypothesis-driven approach in which we allow the model to respond (i.e., make predictions) to stimuli we devise that is of theoretical interest to the investigators. This approach is imported directly from experimental psychology in which the objective is to mechanistically understand cognition. This is a flexible approach that allows the investigator to directly devise stimuli in an intuitive manner which address questions about what the model has learned from the information it was trained on.

To demonstrate the utility of this approach, we explore a data set of essays taken from a series of field studies of a written educational intervention conducted in a North American middle school (Cohen et al., 2009, 2006; Cook et al., 2012). This intervention is designed to mitigate the detrimental effects of stress on student performance. Researchers have demonstrated its effectiveness at boosting the performance of students who are under excessive amounts of stress due to having an identity that is stigmatized in academic settings (e.g. African American, or women in STEM classes). The typical format of the intervention has students select from a list of values (e.g. religion, sports, relationships) and write about why the value they selected is important to them. This values affirmation intervention has reduced the achievement gap between African American and White students in middle school (Cohen et al., 2009), Latino students and White students in mid-

dle school (Sherman et al., 2013), and between male and female students in a college physics class (Miyake et al., 2010), among other examples (reviewed in Cohen and Sherman, 2014).

However, there has been little work exploring the text that students produce while they are writing these essays. We study the justifications that students provide when writing about their values. That is, because the writing prompts instruct students to select a value, and then justify that selection by writing about why it is important, we sought to understand whether these justifications differ between categories of students. In particular, we examine whether female students are more likely than male students to justify their value selection by making reference to others. We also examine whether male students are more likely than female students to justify their value selection by making reference to the self - in particular, types of self-action (Section 3.3). Both of these hypotheses are motivated by previous psychological research. Specifically, women are thought to generally tend toward an interdependent self-construal (Gabriel and Gardner, 1999), which implies a greater emphasis on social relationships (Kuebli and Fivush, 1992; Broderick and Beltz, 1996). Similarly, males are believed to tend toward an independent self-construal, which implies greater emphasis on the self and would suggest that they would be more likely to justify their value selection by referring directly to the self or with a self-action (Cross et al., 2000).

To be able to carry out these hypothesis-driven studies, we proposed an approach that aims to understanding LSTM predictions at a higher level of abstraction (i.e., from the output distribution directly rather than by analyzing hidden states and/or gate activations) (Section 2). In short, we specify a set of inputs that vary along some dimension of theoretical interest, and observe the outputs (probability distributions of classes) that correspond to these inputs. To achieve this, we train the proposed neural network architecture in a supervised way with a dense softmax layer as the last layer at the end of the last time step. During test time, we use this last layer at each time step and record the outputs to understand the sequential class probabilities. We show that our methods lead to more interpretable results in the task of understanding the gender-specific characteristics of text from values affirmation essays.

## 1.1 Related work

**Linguistic Analysis of Values Affirmation** To our knowledge, there are just three other papers directly addressing linguistic data in values-affirmation interventions. Shnabel et al., (2013) hand coded a small sample of values-affirmation essays, looking for whether they contained themes of *social belonging*. Their results found that not only were essays written by writers in the treatment condition more likely to contain such themes, but those writers in the treatment condition who wrote about these topics were also more likely to perform well academically than writers in the treatment condition who did not write about social belonging. This effect was unexpectedly reversed for students who were not under threat - White students in these data. Though this reversal observed for white students could reflect some deeper underlying theoretical insight, it is worth noting that this result is based on a relatively small sample of white students ($n = 186$, approximately half of whom were in the focal condition), and as such could be explained by a lack of power to estimate this type of effect. Taking a similar approach, with a similarly sized dataset, Tibbetts et al. (2016) found that first-generation college students (another group who typically experiences threat in an academic environment) who wrote about independence in the treatment condition performed better academically than those who did not write about this theme. Finally, Riddle et al. (2015) take a more descriptive approach to these data, using topic models to look for topics that distinguished writers of different ethnicities and genders.

**Interpreting Neural Networks** Due to their high-level performance on a number of tasks, there has been some recent work attempting to understand neural networks beyond simple performance metrics. Hermans and Schrauwen (2013) worked on understanding long term dependencies such as detecting parenthesis using recurrent networks by analyzing state activations. Relatedly, Karpathy et al. (2015) used the linux kernel source code dataset to study similar long term dependencies such as in bracketing and line-lengths, incorporating analysis of gate activations during prediction to understand their models. Faruqui et al. (2015) learn sparse word vectors, and show that these vectors can be linked to semantic lexicons and word properties. Additionally, Lei et al. (2016)

uses an encoder-generator framework to automatically generate rationalizations of predictions without the need for manual annotations.

To our knowledge there has been no work that attempts to understand LSTM predictions at a higher level of abstraction (i.e., from the output distributions directly rather than analysis of hidden states and/or gate activations).

This approach borrows directly from standard experimental psychology methods. These methods are designed to test hypotheses about why a decision maker behaves in as observed. Applying these same approaches to understanding LSTM can enable quantitative scientists to explore specific hypotheses in their data, as we demonstrate below.

## 2 Methods

Our aim in this work is to develop a system that allows the applied quantitative scientist to query the network with a specific set of test input (*stimuli*) of his or her choosing that can highlight a question of theoretical interest. In our case, we are interested in understanding textual features of writing that make a values affirmation essay more likely to have been written by a male versus female student in the treatment or control conditions.

### 2.1 Data

Our data come from a series of field studies conducted at a middle school in the northeastern United States with 8 cohorts of students. These studies, described in part elsewhere (Cohen et al., 2009, 2006) were designed to test the effectiveness of a written psychological intervention, called a values-affirmation on reducing the academic achievement gap. At the beginning of their time in middle school, and at several other times throughout their middle school years, students completed a short essay in response to either a treatment or control writing prompt. In the treatment condition, students were typically asked to look at list of values (*athletic ability, art, being smart and getting good grades, creativity, independence, social groups, music, politics, relationships, religion, sense of humor*, or *living in the moment*), and pick one to three that were most important to them and write about why that/those values were important. The control condition was similar, but students were instead asked to pick one to three values that might be important to someone else

and write about why they might be important to someone else. [1] An example of of a treatment and control essay are included below:

> **Treatment:** They are important because I like listening to music and playing my flute. I like having friends and I also enjoy being funny sometimes.

> **Control:** Art might be important to someone else because they might like art, want to go to school to study art to one day become an artist.

In total, the dataset consists of 6224 essays written by 1233 students. Female students wrote 3315 of these essays (of which 1851 were control essays), while male students wrote 2909 (of which 1622 were control essays). For an unrelated project, research assistants manually fixed all gross spelling and grammatical mistakes. The resulting cleaned essays have an average length of 42.8 words (SD = 23.9), and are composed of 5816 distinct tokens.

### 2.2 Neural Network Architecture

Our classification task is to categorize an essay into one of four classes consisting of the four combinations of gender (male versus female) and experimental condition (treatment versus control).

Our NN architectures are depicted in Figures 1 and 2. Figure 1 is the training architecture while figure 2 is the testing architecture. We initialize our network by using the 840B 300d Stanford GloVe word embedding vectors (Pennington et al., 2014) as a bias for words that exist in our data.

**Input Layer:** . All essays are first tokenized, and each word in the essay is mapped to an index in a dictionary. The essays are fed as input as a list of constantly-sized integers to the neural network - shorter essays were padded to create inputs of the same length.

---

[1]The nature of the treatment and control prompts changed somewhat across interventions in order to keep students engaged and to examine the effects of small tweaks of the intervention (e.g. writing about one versus three values).

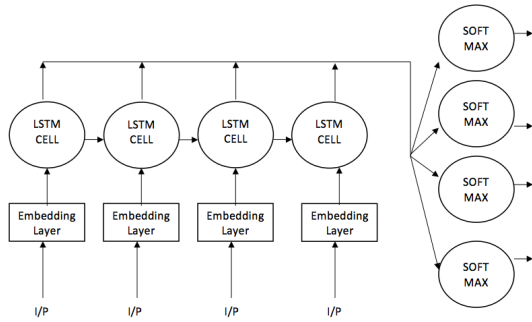

Figure 2: Neural network architecture (testing)

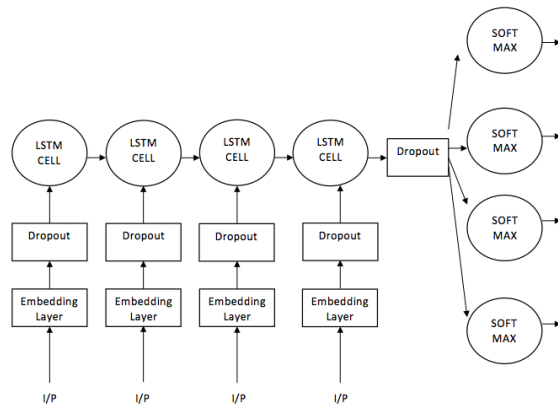

Figure 1: Neural network architecture (training)

**Embedding Layer:** . The input vector is fed to the embedding layer which converts each word into a distributional vector of 300 dimensions. If these vectors are found in the GloVe dataset , they are derived from the corresponding GloVe vector. If not, they are initialized to 0. This layer is trainable. The input token index is mapped to one of the 300 dimensional vectors and the set of indices mapped to the 300 dimensional vectors is sent as the output of this layer.

The output from the embedding layer is of dimension $(T_i, 300)$, where $T$ is the number of tokens in essay $i$. This output is fed to a dropout layer for the purpose of regularization, which drops units with probability .5 during training to avoid overfitting. During test time, no units are dropped.

**LSTM layer:** The output from the dropout layer is fed to an LSTM layer to capture long term and short term dependencies. The state at the last time step of the LSTM layer is sent to a dropout layer

to regularize and then a fully connected layer of dimension 4 (one for each of the output classes) with a softmax activation function to classify to one of the four outputs in terms of a probability distributions.

The loss function used is categorical cross entropy:

$$\mathcal{L}(X,Y) = -\frac{1}{n}\sum_{i=1}^{n} y^{(i)}\ln a(x^{(i)})$$
$$+ (1 - y^{(i)})\ln(1 - a((x^{(i)}))) \quad (1)$$

where $X = \{x^{(1)}, ..., x^{(n)}\}$ is the set of input examples in the training dataset and $Y = \{y^{(1)}, ..., y^{(n)}\}$ is the corresponding set of labels for those input examples and $a(x)$ represents the output of the neural network, given input $x$.

For optimization, we employ rmsprop - Dividing the learning rate for a weight by a running average of the magnitudes of recent gradients for that weight (Tieleman and Hinton, 2012). The metrics used while training are precision, recall, macro f1, and categorical accuracy. The minibatch size used is 256, and the dimension of the hidden state in the LSTM is 50.

At test time, the architecture is changed such that the output at each timestep is fed to the softmax layer to predict the probability of the observed class at that time instant (Figure 2). This modification helps us to understand how specified sequences of text contribute to the classification decision.

The model was constructed and fit using the Keras interface (Chollet, 2015) in Python 2.7 with a Theano backend (Theano Development Team, 2016).

## 3 Results

### 3.1 Model Performance

For all model comparisons, we used a 85/15 train/test split to evaluate performance. Our baseline model - a linear support vector classifier was trained to identify the four gender and condition categories (male & control, male & treatment, female & control, female & treatment) using features imported from previous work (Riddle et al., 2015). Specifically, this model included a tf-idf language model with standard preprocessing steps (lemmatized, stop words removed, words with document frequency ¡ 4 removed), and a 50-topic

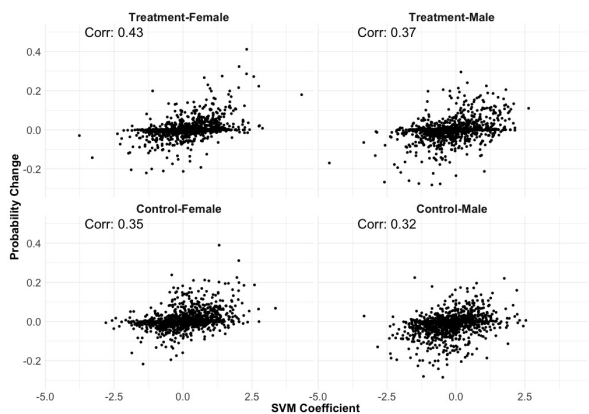

Figure 3: Correlations between SVM coefficients and LSTM probability changes

lda topic model. The linear support vector classifier was trained using 10-fold cross-validation, with each fold using 15% of the training data as a validation set. This model achieved a macro f1 score of .53 on the final test set. In contrast, the LSTM model, which was validated during training on a held-out 15% performed better, achieving a macro f1 score of .60 on the final test set.

## 3.2 Validating the Interpretations

As described above, our model emits class probabilities at each word. In order to establish that these probabilities are valid ways to interpret the model, we compare them with coefficients from our baseline classifier. Because the probability change for the LSTM is context dependent, each token is possibly associated with multiple changes in probability. Thus, for each token, we obtained the average probability change for each class. As seen in figure 3, there are reasonably strong correlations (pearson's correlation coefficients between .32 and .43) between a token's coefficient from the SVM model and that token's average probability change from the LSTM. This provides some evidence that the probability shifts for a token measure the same underlying construct as SVM coefficients.

**Example Interpretation:** In Figure 4, we show an illustrative example of model interpretation. In this example, we use an example essay written by a female student in the treatment condition:

> They are important because I like listening to music and playing my flute. I

like having friends and I also enjoy being funny sometimes.

There are a number of noteworthy points in the example essay where the probabilities shift in a way that would be expected based on the nature of the writing prompt and the interests of male and female American middle schoolers. For instance, after the first five words, the model is beginning to predict that the essay is from the treatment condition. The first five words, *They are important because I* are more typical for writing in response to the treatment prompt (*write about why the values are important to you*) as opposed to the control prompt (*write about why the values are important to someone else*). Another noteworthy shift here is when the model encounters the word *flute*, an instrument that is relatively gendered in American culture (i.e. it is more strongly associated with women than with men). At this point, the model is positive that this is an essay from the treatment condition, written by a female student.

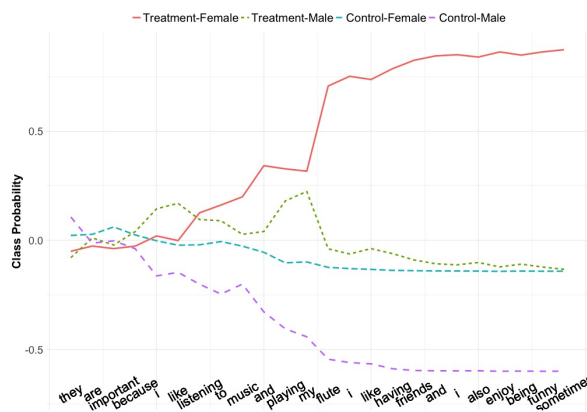

Figure 4: Class probability changes in an example essay

## 3.3 Making Psychological Queries

Having validated the approach to interpreting the model and shown an example case, we next turn to asking richer questions about our data. In particular, we present evidence that suggests that male writers and female writers use slightly different justifications for selecting values. Given that the writing prompt for the treatment condition asks students to *"write about why* [their chosen value] *is important to you"*, these justifications can tell us something about the underlying needs and motivations for some group of writers - in this case, male versus female students.

To explore this question, we refit the LSTM model on the entire dataset. After adjusting to the model to obtain class probabilities through the essay as described above, we adapted methods from experimental psychology to query what the model has learned. In psychological experiments, human participants are often given linguistic items that differ along some key dimension of interest while the experimenter records the responses to these items. We took this approach to investigating the model, with the responses consisting of the sequential probabilities.

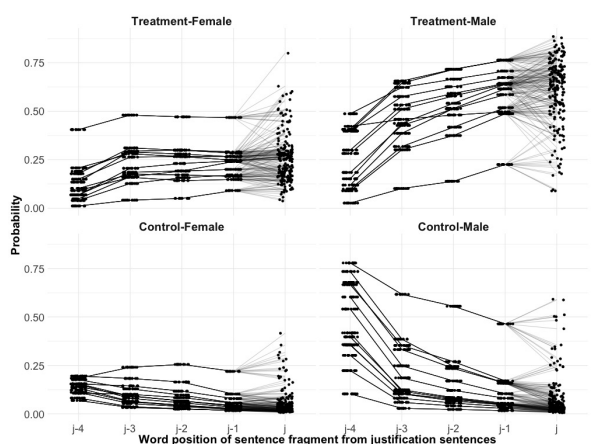

Figure 5: Sequential probabilities for the last five words in the other-justified test sentences. The final word (*justification*) is marked as j. Each line is an individual sentence fragment's sequential probabilities.

The dimension of interest that was systematically varied was the type of justification used in a stereotypical treatment essay. Specifically, we fed the model a series of sentence segments taking the form of "*V is important to me because J*", where the *V* is one of the values that students could have selected (e.g. athletic ability, getting good grades, etc), and *J* was a series of bigrams that was one of two types - either self-focused justifications, or other-focused justifications. As described in the introduction, our expectation is that female students will be more likely to use other-focused justifications in which they describe their value in relation to other people because this reflects a greater degree of interdependent self-construal (e.g. a female student would be more likely to write "*Art is important to me because my mom...*"). For male students, we expect them to be more likely to use self-focused justi-

fications, reflecting a greater degree of independent self-construal. Such a justification would take them form of their explaining a value by making some reference directly to the self doing, thinking, or performing something (e.g. a male student might be more likely to write "*Art is important to me because I would...*").

Table 1 displays the values and justifications that were used for the experiments. Using these combinations, we generated 195 other-focused sentence segments and 195 self-focused sentence segments for testing. This methodological approach can tell us whether one of these classes is more or less likely to use other-focused versus self-focused justifications.

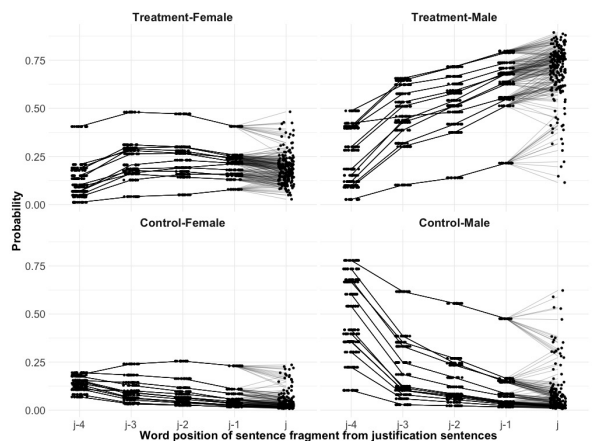

Figure 6: Sequential probabilities for the last five words in the self-justified test sentences. The final word (*justification*) is marked as j. Each line is an individual sentence fragment's sequential probabilities.

Figures 5 and 6 displays the sequence of probabilities for the last five words in the generated sentence fragments. As expected, the probabilities for a given value are identical for a given class until the justification word is encountered, at which points the probabilities diverge. Because these probabilities are compositional in nature (i.e. there are $D$ alternative probabilities in the vector of any time point and these alternatives must sum to 1), we first transformed the probabilities for each sentence fragment at each time step according to the centered log-ratio transformation (Aitchison, 1986). This transformation is of the

| Value | other-focused | self-focused |
|---|---|---|
| Athletic ability | my friends | I want |
| Art | my family | I need |
| Being smart | my grandparents | I will |
| Getting good grades | my parents | I have |
| Creativity | my teacher | I can |
| Independence | my friend | I feel |
| Social groups | my dad | I should |
| Music | my mom | I would |
| Politics | my sister | I hope |
| Relationships | my brother | I am |
| Religion | my mother | I get |
| Sense of Humor | my father | I might |
| Living in the moment | my cousin | I use |
| - | my aunt | I like |
| - | my uncle | I take |

Table 1: Combinations of values and justifications (other-focused versus self-focused) used in the test sentence segments

form:

$$log(\frac{x_1}{g(x)}, \frac{x_2}{g(x)}, ... \frac{x_D}{g(x)}) \quad (2)$$

Where $x_D$ are the class probabilities, and $g(x)$ is the geometric mean of the probabilities.

To quantify the effect of these two different justifications, we fit two multilevel bayesian model to the centered log-ratio transformed LSTM output (Baayen et al., 2008) - one model for other-focused sentences and one model for self-focused sentences. In mixed-effects terminology, these model had fixed effects terms for class, time, and their interaction, and a three fully crossed random effects - one for sentence fragment, one for the value being justified, and one for the different justifications. Each of these crossed random effects has its own vector of intercept and time coefficients. These types of models are a preferred approach for the analysis of this type of psycholinguistic data, or any other type of data in which the experimenter is recording responses from a participant (or in our case, an LSTM network) to stimuli that are sampled from some larger class (Baayen et al., 2008; Westfall et al., 2016; Judd et al., 2017). All priors were set to a standard normal distribution. The bayesian model was fit using Stan (Carpenter et al., 2016) and the Rstanarm interface (Gabry and Goodrich, 2016).

Taking draws from the posterior distribution, we obtained the difference in probability distribution for word $J$ and word $J - 1$ for each class. Because our hypotheses were primarily about how male and female students would justify their selection in values, we then subtracted these two difference distribution, yielding a posterior distribution that quantifies the degree to which the lstm model believes that female students are more likely to use a given type of justification. More formally:

$$\delta_m = X_{mj} - X_{mp} \quad (3a)$$
$$\delta_f = X_{fj} - X_{fp} \quad (3b)$$
$$F_{pref} = \delta_f - \delta_m \quad (3c)$$

where $X$ designates the posterior distribution estimated directly in the bayesian model, the subscripts $m$ and $f$ reference male and female students, respectively, and the subscripts $j$ and $p$ reference the justification word and the word preceding the justification word, respectively. $X_{mj}$ is the posterior distribution for the justification word $J$ for males, $X$ The intuition is that we are interested in if the justification word $J$ differentially increased probability for female students over male students or vice versa.

For other-focused justification, the analysis indicates a weak tendency for these types of justifications to be used more often by women (mean of $F_{pref} = .06$, 95% highest density interval (HDI) $= [-.11, .22]$), with the posterior distribution indicating that $74\%$ of the posterior dis-

tribution is consistent with an effect of this direction. In real terms, the overall probability of the treatment-female class increased from a mean of $.24 (SD = .09)$ at the penultimate word to a mean of $.26 (SD = .13)$ after seeing the justification word, while the overall probability of the male-treatment class increased a very small amount from $.60 (SD = .14)$ to $.61 (SD = .17)$ for the justification word. The analysis of self-focused justifications is in the opposite direction, with the effect suggesting stronger support for the hypothesis. Namely, the LSTM model seems to have learned that male students may be more likely to justify their values with language that reflects a self-focus (mean of $F_{pref} = -.23$, 95% HDI $= [-.39, .07]$, with the posterior distribution indicating that 99.8% of the posterior distribution is consistent with an effect of this direction. Again, in real terms, the overall probability of the female-treatment class decreased from a mean of $.21, (SD = .08)$ at the penultimate word to a mean of $.19, (SD = .08)$ at the justification word, while the overall probability of the male-treatment class increased from a probability of $.63, (SD = .15)$ to a mean of $.69, (SD = .14)$.

## 4 Conclusion

We have described a novel way to explore neural networks. By treating the model as a input-output device and carefully crafting stimuli that systematically varies along a dimension of interest, we can map systematic differences in the output to the stimulus input. Doing so allows us to understand whether the model has learned about this dimension, and consequently, whether there is information of this type in the training data, even if the same language does not appear verbatim.

Our work also extends literature in psychology, as this is the first instance in which the justifications of values-affirmation essays were explored. We find evidence largely consistent with expected gender differences as a function of theories of gendered self-construal (Gabriel and Gardner, 1999; Kuebli and Fivush, 1992; Broderick and Beltz, 1996). Future work should disentangle the degree to which these justifications influence the effectiveness of the intervention on academic outcomes.

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
