# Peer review of "Interpreting Neural Networks to Understand Written Justifications in Values-Affirmation Essays"

_ACL 2017 — decision unknown_

[Official Review · Reviewer 1 · rating 2 · confidence 4]
soundness 3 · originality 4 · clarity 4 · impact 3 · substance 2 · appropriateness 5 · meaningful comparison 5 · presentation format Poster

- Strengths: this paper addresses (in part) the problem of interpreting Long
Short-Term Memory (LSTM) neural network models trained to categorize written
justifications in values-affirmation essays. This is definitely an interesting
research question. To do so, the authors want to rely on approaches that have
are standard in experimental psychology. Furthermore, the authors also aim at
validating sociological assumptions via this study.

- Weaknesses: one of the main weaknesses of the paper lies in the fact that the
goals are not clear enough. One overall, ambitious goal put forward by the
authors is to use approaches from experimental psychology to interpret LSTMs.
However, no clear methodology to do so is presented in the paper. On the other
hand, if the goal is to validate sociological assumptions, then one should do
so by studying the relationships between gender markers and the written
justifications, independently on any model. The claim that "expected gender
differences (are) a function of theories of gendered self-construal" is not
proven in the study.

- General Discussion: if the study is interesting, it suffers from several weak
arguments. First of all, the fact that the probability shift of a token in the
LSTM network are correlated with the corresponding SVM coefficients is no proof
that "these probabilities are valid ways to interpret the model". Indeed, (a)
SVM coefficients only reveal part of what is happening in the decision function
of an SVM classifie and (b) it is not because one coefficient provides an
interpretation in one model that a correlated coefficient provides an
explanation in another model. Furthermore, the correlation coefficients are not
that high, so that the point put forward is not really backed up.

As mentioned before, another problem lies in the fact that the authors seem to
hesitate between two goals. It would be better to clearly state one goal and
develop it. Concerning the relation to experimental psychology, which is a
priori an important part of the paper, it would be interesting to develop and
better explain the multilevel bayesian models used to quantify the gender-based
self-construal assumptions. It is very difficult to assess whether the
methodology used here is really appropriate without more details. As this is an
important aspect of the method, it should be further detailed.

[Official Review · Reviewer 2 · rating 3 · confidence 1]
soundness 3 · originality 4 · clarity 3 · impact 3 · substance 4 · appropriateness 5 · meaningful comparison 5 · presentation format Poster

- Strengths:
The paper is thoroughly written and discusses its approach compared to
other approaches. The authors are aware that their findings are somewhat
limited regarding the mean F values.

- Weaknesses:
Some minor orthographical mistakes and some repetive clauses. In general the
paper would benefit if the sections 1 and 2 would be shortened to allow the
extension of sections 3 and 4.
The main goal is not laid out clearly enough, which may be a result of the
ambivalence of the paper's goals.

- General Discussion:
Table 1 should only be one column wide, while the figures, especially 3, 5, and
6 would greatly benefit from a two column width.
The paper was not very easy to understand during first read. Major improvements
could be achieved by straightening up the content.